# Intravesical Chemotherapy after Radical Nephroureterectomy for Primary Upper Tract Urothelial Carcinoma: A Systematic Review and Network Meta-Analysis

**DOI:** 10.3390/jcm8071059

**Published:** 2019-07-19

**Authors:** Sang Hyun Yoo, Chang Wook Jeong, Cheol Kwak, Hyeon Hoe Kim, Min Soo Choo, Ja Hyeon Ku

**Affiliations:** 1Department of Urology, Seoul National University Hospital, Seoul 03080, Korea,; 2Department of Urology, Hallym University Dongtan Sacred Heart Hospital, Hwaseong 18450, Korea

**Keywords:** nephroureterectomy, upper tract urothelial carcinoma, bladder cancer, intravesical chemotherapy, meta-analysis

## Abstract

The aim of this study was to determine the prophylactic effect of intravesical chemotherapy. Furthermore, it aimed to compare the efficacy of regimens on the prevention of bladder recurrence, after nephroureterectomy, for upper tract urothelial carcinoma by systematic review and network meta-analysis. A comprehensive literature search was conducted to search for studies published before 22 December 2016 using PubMed, Embase, and Scopus. All studies comparing nephroureterectomy alone with prophylactic intravesical chemotherapy after nephroureterectomy were included. The primary outcome was intravesical recurrence-free survival rate. In addition, we conducted indirect comparisons among regimens using network meta-analysis, as well as three randomized controlled trials (RCTs) on multicenter setting, and one large retrospective study with a total of 532 patients were analyzed. The pooled hazard ratio (HR) of bladder recurrence was 0.54 (95% CI: 0.38–0.76) in intravesical instillation patients. On network meta-analysis, pirarubicin was ranked the most effective regimen, while maintenance therapy of mitomycin C (MMC) with Ara-C and induction therapy of MMC were ranked as the second and third most effective regimens, respectively. Our study demonstrates that intravesical chemotherapy can prevent bladder recurrence in patients with upper tract urothelial carcinoma after nephroureterectomy. It also suggests that a single instillation of pirarubicin is the most efficacious intravesical regimen.

## 1. Introduction 

Urothelial cell carcinoma is known as the fifth most common cancer [1]. Although upper tract urothelial carcinoma is a disease that accounts for 5–10% of urothelial carcinoma, its incidence has been reported to be as high as 20–25% in Asia due to herbal medicines [2,3]. The standard treatment of upper tract urothelial carcinoma is nephroureterectomy with bladder cuff excision [4]. However, recurrence rate of bladder cancer is 22–47% after standard radical surgery [5]. Recurrence rate in the contralateral upper tract is about 2–6% [6,7,8]. Prevention is necessary to reduce inconvenience and medical cost associated with additional bladder surgery and preserve the opposite renal function. 

It has been reported that intravesical chemotherapy after transurethral resection of the bladder (TURB) can reduce tumor recurrence by 26–40% [9]. However, there is still no consensus for standard prophylactic intravesical chemotherapy in upper tract urothelial carcinoma. Some studies have reported the usefulness of epirubicin, mitomycin C (MMC), and pirarubicin [9,10,11]. However, they remain controversial. Only the European Association of Urology (EAU) guideline advocates prophylactic adjuvant intravesical chemotherapy [12]. Furthermore, there is no comparative study to determine which regimen is the best to use.

When comparing effects between two groups based on a systematic review, it is most ideal if there is a direct comparison of randomized clinical trials between the two groups of interest. However, there are many cases where there is no direct comparison of randomized controlled trials (RCT). In such cases, network meta-analysis method such as indirect comparison and mixed comparison can be utilized. Network meta-analysis and multiple treatment comparisons of randomized controlled trials have been introduced to facilitate the indirect comparison of multiple interventions not studied in head-to-head studies [13]. In addition, network meta-analyses allow the visualization of a larger amount of evidence, estimation of the relative effectiveness of all interventions, and rank ordering of interventions [14,15].

The objective of this study was to determine the efficacy of intravesical chemotherapy in preventing bladder cancer recurrence after nephroureterectomy. In addition, the study aimed to find the most effective drugs for upper tract urothelial carcinoma by systematic review and network meta-analysis. 

## 2. Materials and Methods 

The meta-analysis was performed in accordance with Cochrane Collaboration and Preferred Reporting Items for Systematic Review and Meta-analysis (PRISMA) statement [16].

### 2.1. Search Strategy 

A comprehensive literature search was conducted to identify studies published before 22 December 2016 using PubMed, Embase, and Scopus. We only reviewed studies published in English language. The following keywords were used: (upper urinary tract) AND (urothelial carcinoma) AND (intravesical) AND (bladder). References of retrieved articles were also examined to identify other potentially eligible studies for inclusion if they were not included in the initial automated search. Article selection was made by two independent evaluators (J.H.K., C.W.J.). All discrepancies between the two were resolved by discussion and consensus. 

### 2.2. Eligible Criteria

We defined study eligibility according to predefined selection criteria [16].

Population: Patients who underwent radical nephroureterectomy for primary upper tract urothelial carcinoma. 

Interventions: Intravesical chemotherapy.

Comparators: Radical nephroureterectomy only. 

Outcomes: Intravesical recurrence-fee survival rates.

Study design: Prospective or retrospective.

We used strict inclusion and exclusion criteria to limit heterogeneity across studies. 

The following criteria for eligibility among studies were set before collecting articles:(1)Articles included patients with primary upper tract urothelial carcinoma.(2)Articles compared intravesical recurrence with or without intravesical chemotherapy.(3)Articles reported intravesical-recurrence free survival rates after radical nephroureterectomy.

Accordingly, the following exclusion criteria were used:(1)Non-human study.(2)Review articles, letters, editorial comments, and case reports.(3)Articles without sufficient data to acquire hazard ratio (HR) or standard error (SE).(4)Articles including conservative surgery instead of radical surgery, other neoadjuvant or adjuvant treatment, metastatic disease, or non-urothelial carcinoma.

When multiple publications from the same group based on similar patients were available, we included the publication with the largest number of cases and the most applicable information. 

### 2.3. Data Extraction

Two authors (C.K. and H.H.K.) completed an independent review of 350 articles. A total of 336 articles were excluded after examining titles and abstracts. Full texts of 14 articles were evaluated. In accordance with all previously mentioned inclusion criteria, four studies were finally enrolled in this systematic review and meta-analysis [7,9,10,11]. Any discrepancy between the two authors was resolved by discussion. PRISMA flow chart depicting the process for systematic literature search and selection of studies is shown in Figure 1.

Separate data tables were independently made to extract all relevant data from tests, tables, and figures of each included study. The following information was obtained: author’s name, year of publication, geographic location, period of recruitment, study design, number of patients, median age, gender, tumor characteristics, intravesical chemotherapy delivered, instillation duration, and intravesical recurrence-free survival. Intravesical recurrence-free survival was defined as the interval between surgery and urothelial carcinoma in the bladder. 

### 2.4. Statistical Analysis

A direct meta-analysis was conducted and a random effects HR was calculated according to DerSimonian and Laird [17]. We extracted or estimated the logarithm of HR (log[HR]) and its variance. Thusly, we calculated the HRs and the corresponding 95% confidence intervals (CIs) to assess the effect of each chemotherapy regimen on the outcomes. The exchangeability was assessed by examining heterogeneity in each head-to-head comparison. Cochrane Q test and *I*^2^ test were used to assess between-study heterogeneity of HRs. A p value of less than 0.05 for the Cochran Q test or an *I*^2^ statistic >50% indicated the presence of significant heterogeneity across selected studies [17,18]. Inverted funnel plots, the Begg test (rank correlation analysis), and the Egger test (linear regression analysis) were used to evaluate publication bias [19,20]. Statistically significant publication bias was considered when a p value was less than 0.05 for Egger or Begg test. 

For indirect comparisons among regimens, a Bayesian random effects model was used using Markov chain Monte Carlo methods [21]. The selection of a fixed or random effects model for reported outcomes was based on model fit criteria (Deviance Information Criteria, DIC) penalizing greater model complexity [22]. We incorporated all data presentations in a single analysis using methods described by Woods et al. [23]. The median of the posterior distribution was used as a point estimate for treatment effect. We modeled binary outcomes in every treatment group of every study and specified relations among odds ratios with 95% credible interval (CrI) across studies to make different comparisons. In the presence of minimally informative priors, CrIs can be interpreted similarly to conventional CIs. Each analysis was based on noninformative priors for effect sizes and precision. 

We also examined inconsistency between direct and indirect estimates using a modified back-calculation approach [24]. Quality of model was examined by inspecting convergence using Gelman-Rubin-Brooks plots, assessing autocorrelation between iterations of the Markov chain, and determining whether Monte Carlo (MC) error was less than 5% of posterior standard deviation.

Meta-analysis was carried out using Review Manager v.5.1 (The Nordic Cochrane Center, The Cochrane Collaboration, Copenhagen, Denmark, 2008) and R 2.13.0 (R development Core Team, Vienna, Austria, http://www.R-project.org). Bayesian framework meta-analyses were performed using WinBUGS 1.4 (MRC Biostatistics Unit, Cambridge, UK). Two-sided p value of less than 0.05 was considered statistically significant except for heterogeneity test, in which a one-sided p value of less than 0.1 was used. 

## 3. Results 

### 3.1. Overview of Studies Included

Table 1 and Table 2 show individual data on characteristics of the four included studies [7,9,10,11] and patient population. Studies were published between 2001 and 2013. Patient recruitment period ranged from 1985 to 2008. Three studies were prospective controlled trials [9,10,11]. Mitocycin-C (MMC) plus cytarabine (cytosine arabinoside, Ara-C), epirubicin, and MMC plus prirubicin were used in each trial. A single instillation was given in two studies [9,11], while the other two studies used repeated instillations over time [7,10]. One retrospective cohort study was a comparison of epirubicin, MMC, and the control. There was no difference in baseline characteristics between the three groups. All studies compared no instillation as the control. Total sample size was 532, with a mean of 133 (range: 25–239 patients). Two datasets included <100 patients [10,11], while two data sets had enrolled ≥100 patients [7,9]. The overall risk of bias in the included studies is presented graphically in Figure 2 and Figure 3.

### 3.2. Pair-Wise Meta-Analysis

According to a priori assumption about the likelihood for heterogeneity between primary studies, pooled HR estimate of each study was calculated using the random effect model. Pooled analysis of intravesical recurrence-free survival indicated that intravesical chemotherapy after radical nephroureterectomy had better intravesical recurrence-free survival when compared to radical nephroureterectomy only (HR: 0.54; 95% CI: 0.38–0.76; *p =* 0.0004). There was no significant heterogeneity among these studies (*p =* 0.43; *I*^2^ = 0%) (Figure 4A). Three prospective controlled trials reported analysis results for comparison, showing that intravesical chemotherapy after radical nephroureterectomy was significantly associated with better intravesical recurrence-free survival (HR: 0.46; 95% CI: 0.27–0.78; *p =* 0.0004). Test of inconsistency excluded significant heterogeneity (*p =* 0.33; *I*^2^ = 11%) (Figure 4B). Funnel plots demonstrated no evidence of remarkable asymmetry. The Begg test indicated no significant (*p >* 0.05) publication bias among these studies. However, the use of the Egger test for these four studies demonstrated a significant (*p =* 0.044) publication bias.

### 3.3. Bayesian Framework Network Meta-Analysis

Figure 5 illustrates networks of all studies (Figure 5A) and three prospective controlled trials (Figure 5B) included according to comparisons of different regimens. Nodes in a network that are not well-connected should be interpreted with caution.

Figure 6 shows results of network meta-analysis. When radical nephroureterectomy only was considered as the reference for comparison, MMC 10 mg induction (HR: 0.49; 95% CrIs: 0.25–0.98) and pirarubicin 30 mg single instillation (HR: 0.26; 95% CrI: 0.08–0.91) were associated with statistically significant better intravesical recurrence-free survival (Figure 6A). For other regimens, 95% CrIs overlapped the null effect line. We also performed a subgroup analysis, including three prospective controlled trials. When no intravesical chemotherapy was considered as the reference for comparison, pirarubicin 30 mg single instillation (HR: 0.26; 95% CrI: 0.08–0.92) was associated with significantly better intravesical recurrence-free survival (Figure 6B). 

Rankings of six different treatment strategies (including control) in terms of intravesical recurrence-free survival are provided in Table 3. A single instillation of pirarubicin 30 mg was ranked the best, while maintenance therapy of MMC 20 mg + Ara-C 20 mg and induction therapy of MMC 10 mg were ranked as the second and third most effective regimens, respectively. In subgroup analysis that included three prospective controlled trials, single instillation of pirarubicin 30 mg usually ranked the best while maintenance therapy of MMC 20 mg + Ara-C 20 mg had a high probability of being ranked the second best. 

## 4. Discussion

In this study, we confirmed that intravesical chemotherapy after nephroureterectomy was effective in reducing bladder cancer recurrence through a systematic review and pairwise meta-analysis. In addition, a network meta-analysis revealed that pirarubicin was the most effective regimen among intravesical chemotherapy strategies. To the best of our knowledge, this is the first study that compares and ranks six different intravesical chemotherapy strategies after nephroureterectomy using network meta-analysis based on a Bayesian random effects model.

The pathogenesis of intravesical recurrence after radical nephroureterectomy for upper tract urothelial carcinoma remains unclear. Multifocal nature is a traditional characteristic of urothelial carcinoma [25]. This has been explained by two hypotheses: Monoclonality hypothesis and field-cancerization hypothesis. The monoclonal hypothesis suggests that intravesical recurrence is a secondary implantation by descendant intraluminal seeding or migration through the urothelial lining [26,27]. The field-cancerization hypothesis suggests that the entire urothelium is exposed to common carcinogenic insults while multifocal tumors are subsequently developed from independent malignant transformation [28,29]. Results of recent genetic studies suggest that both mechanisms might be involved in the development of bladder cancer following previous upper tract urothelial carcinoma [30,31].

Intravesical chemotherapy has been proven to be effective [32,33]. It is widely used for the treatment of bladder tumors and it is considered a standard therapy in bladder tumor management [34]. However, its effectiveness after nephroureterectomy for bladder tumor recurrence remains controversial. The mechanism of action of intravesical instillation therapy involves the delivery of high concentrations of an anticancer drug locally within the bladder without causing general toxicity [35]. Theoretically, it might also be preventive after nephroureterectomy by potentially destroying viable seeding cells from an upper tract or proliferating metachronous tumors in the bladder [2]. In the first RCT performed by Sakamoto et al. [10] on MMC, only a trend was observed. In the study of O’Brien et al. [9], MMC was statistically significant in per-protocol analysis, but not in intend-to-treatment analysis. In a long-term retrospective study by Wu et al. [7], only MMC showed a significant difference in tumor recurrence whereas epirubicin showed no statistical significance. Recently, Ito et al. [11] confirmed the effectiveness of pirarubicin through RCT. We demonstrated that prophylactic intravesical instillation could prevent bladder tumor recurrence after nephroureterectomy for upper tract urothelial carcinoma. 

In this study, pirarubicin was found to be the most effective drug to prevent bladder recurrence. Pirarubicin is more lipophilic with a higher molecular weight than MMC and epirubicin [36]. To maximize exposure, an ideal drug of intravesical chemotherapy should be able to rapidly penetrate to the urothelium instead of being rapidly absorbed into systemic circulation [35]. Lipid solubility is a key determinant of drug penetration across the urothelium [35]. Systemic uptake of any drug administered intravesically is dependent on its molecular weight [35]. Therefore, pharmacokinetic properties of pirarubicin may be considered as more appropriate for intravesical chemotherapy than other drugs. In addition, the observed clinical difference might be explained by the mechanism of action. Anthracyclines such as pirarubicin and epirubicin are not cycle specific. However, MMC, as an anthracyclic-based drug, might be more effective for direct cytoablation with a relatively short period of dwell time [37].

The ideal time point of initiating intravesical chemotherapy instillation after nephroureterectomy has not been standardized yet. Intravesical chemotherapy after TURB immediately after surgery is recommended in guidelines [12,32]. Theoretically, it is best to do instillation immediately postoperative due to implantation of tumor cells after nephroureterectomy or residual without visible disease. However, there is a risk of extravasation after nephroureterectomy with bladder cuff excision. There were three randomized controlled trails for post-nephroureterectomy intravesical chemotherapy, each with different instillation time. In the case of pirarubicin which was proven to be the most effective, patients received a single instillation within 48 hours after nephroureterectomy and retained for 30 minutes [11]. MMC plus Ara-C which was confirmed to be the second most effective was initiated 1–2 weeks after surgery and patients were instructed to refrain from voiding for 2 h after the instillation [10]. Epirubicin was initiated within two weeks after surgery and patients retained the solution for at least one hour [3]. These patients received epirubicin maintenance instillation after the initial 6 to 8 treatments. Therefore, it can be said that earlier instillation time after surgery can lead to better effect.

There are several factors which reported to have a significant impact on intravesical recurrence, including stage, size, grade, and multifocality of the tumor, patients’ history, gender, preoperative renal function, hydronephrosis, cytology, surgical methods, distal ureter management, and surgical margin status [38]. In this study, we have not adjusted these important risk factors. This could be a major drawback of most meta-analysis as well as our study. However, in this study, we included only well-organized and well-conducted RCTs of multicenter settings and a large-scale retrospective study, to minimize these inherent limitations. Therefore, these covariates would not have had a great effect on the conclusion. In addition, it is another limitation to deal with only the impact of the type of agent. Factors affecting the outcome of intravesical chemotherapy include not only the type of agent, but also the characteristics of the disease (previous history of bladder cancer, cis, or type of distal ureter management) and/or the time of administration. Studies on these other factors have yet to be completed at the present time. We hope that this study might be a starting point for other studies.

## 5. Conclusions

Our results confirmed that intravesical chemotherapy could reduce bladder recurrence after nephroureterectomy for upper urinary tract urothelial carcinoma. By comparisons, using network meta-analysis of RCTs, single instillation of pirarubicin within 48 hours after nephroureterectomy was identified as the most efficacious regimen. The maintenance therapy of MMC with Ara-C initiated 1–2 weeks after surgery seemed to be the second most efficacious regimen. Although head-to-head RCTs directly comparing drug efficacies, for more definitive evidence, are still lacking, our study can help the decision making process for selecting regimens of intravesical chemotherapy for patients with upper tract urothelial carcinoma. In addition, we found that the timing of instillation, as well as the regimen of chemotherapy, could be an important factor.

## Figures and Tables

**Figure 1 jcm-08-01059-f001:**
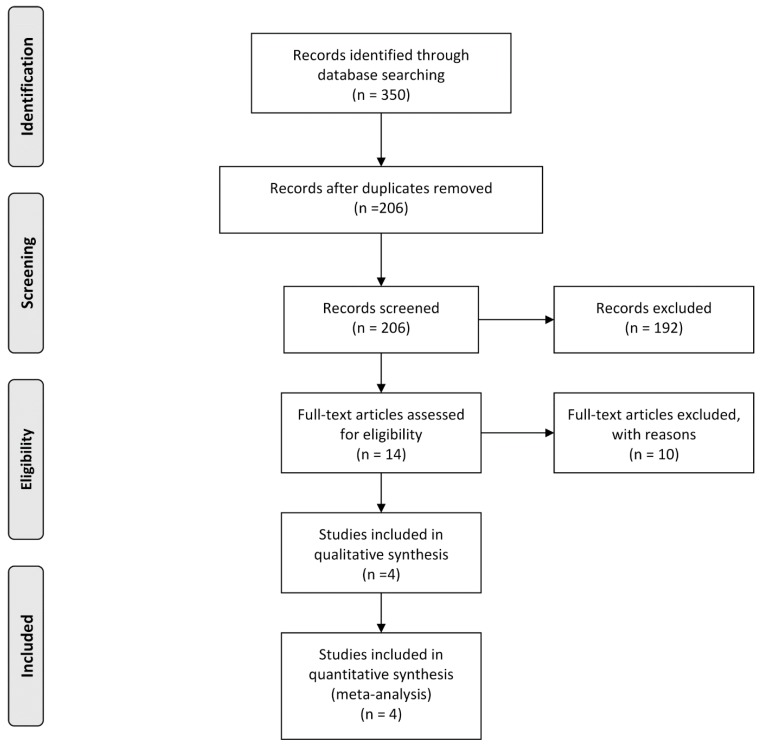
A flow chart showing literature search approach used in this meta-analysis.

**Figure 2 jcm-08-01059-f002:**
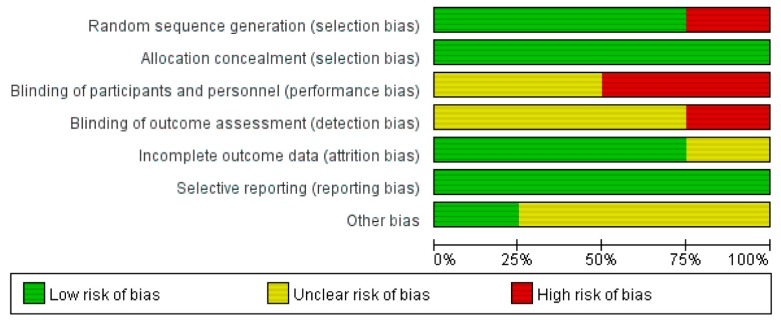
Risk of bias graph: review authors’ judgments about each risk of item presented as percentages across all included studies.

**Figure 3 jcm-08-01059-f003:**
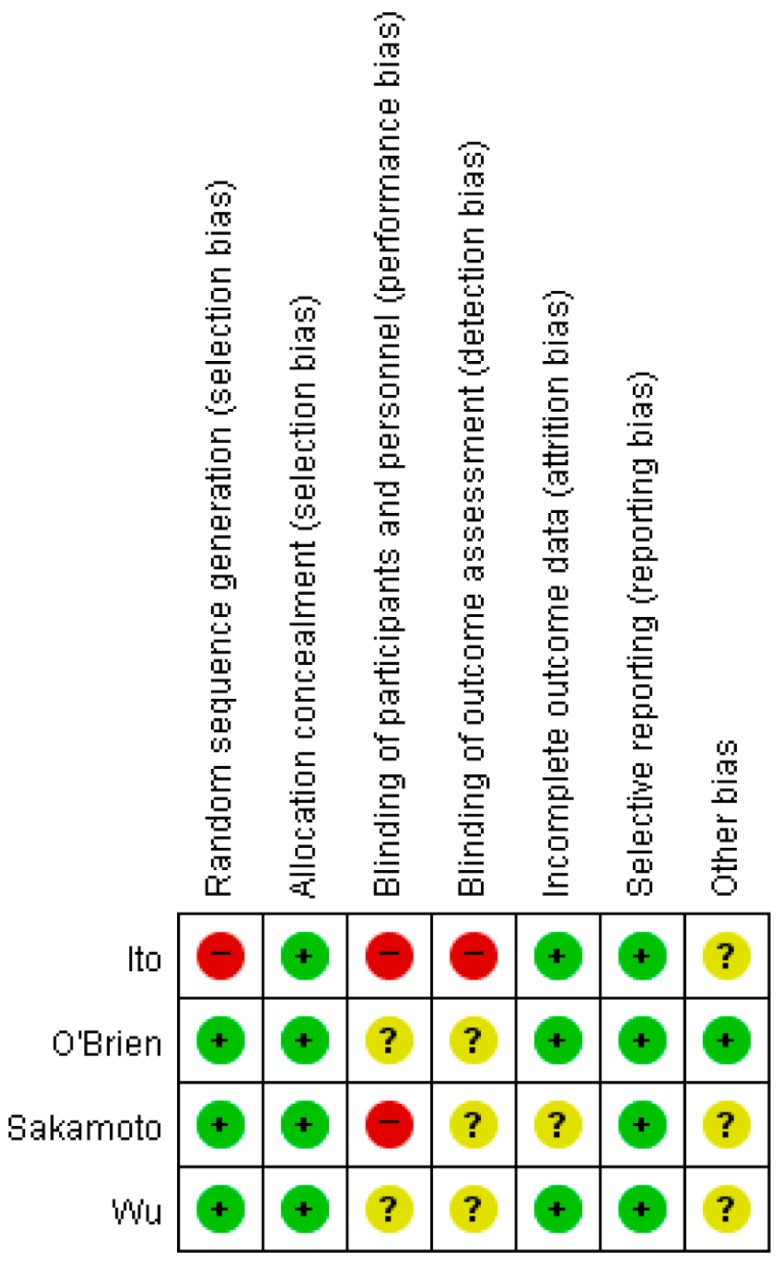
Risk of bias summary: review authors’ judgments about each risk of bias item for each included study.

**Figure 4 jcm-08-01059-f004:**
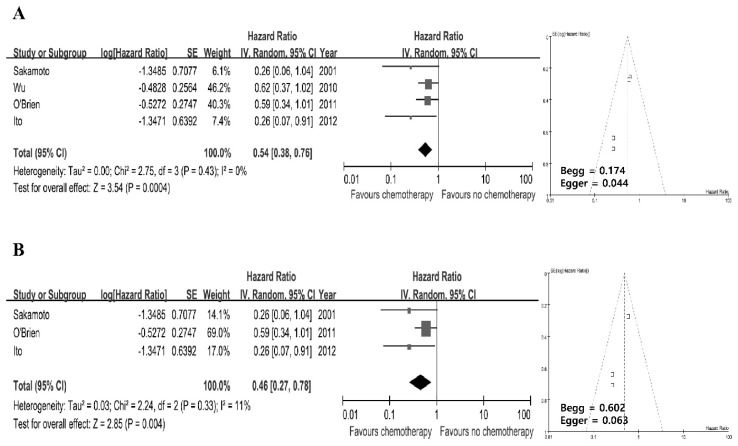
Forest plots (Left). Horizontal lines correspond to study-specific hazard ration and 95% CI. The area of the square reflects study-specific weight. The diamond represents pooled results of hazard ratio and 95% CI. Begg funnel plots for publication bias test are shown (Right). Each point represents a separate study for the indicated association. Vertical line represents mean effects size. (**A**) All studies; (**B**) Prospective controlled studies.

**Figure 5 jcm-08-01059-f005:**
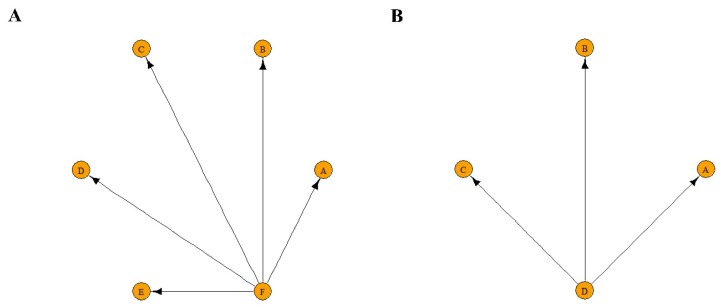
Network geometry of connected network of studies. Circles represent the regimen as a node in the network. Lines represent direct comparisons using studies. Each regimen was compared based on the effect of the control (no instillation). (**A**) All studies. A: Mitomycin C (MMC) 20 mg + Ara-C 200 mg (maintenance). B: MMC 10 mg (induction). C: Epirubicin 20 mg (induction). D: MMC 40 mg (single instillation). E: Pirarubicin 30 mg (single instillation). F: Control (no instillation). (**B**) Prospective controlled trials. A: MMC 20 mg + Ara-C 200 mg (maintenance). B: MMC 40 mg (single instillation). C: Pirarubicin 30 mg (single instillation). D: Control (no instillation).

**Figure 6 jcm-08-01059-f006:**
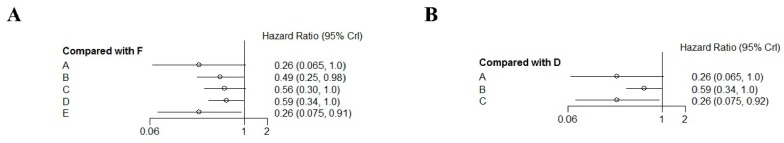
Hazard ratio and 95% credible intervals relative to control. HR < 1 means the regimen is better compared to the control. (**A**) All studies. A: MMC 20 mg + Ara-C 200 mg (maintenance). B: MMC 10 mg (induction). C: Epirubicin 20 mg (induction). D: MMC 40 mg (single instillation). E: Pirarubicin 30 mg (single instillation). F: Control. (**B**) Prospective controlled trials. A: MMC 20 mg + Ara-C 200 mg (maintenance). B: MMC 40 mg (single instillation). C: Pirarubicin 30 mg (single instillation). D: Control.

**Table 1 jcm-08-01059-t001:** Characteristics of eligible studies.

	Year	Country	Study Type	No. of Center	Recruitment Period	No. of Patients (Chemotherapy/Control)	Chemotherapy	Duration
Sakamoto [10]	2001	Japan	Prospective	Multicenter	1993–1996	25 (13/12)	MMC 20 mg and Ara-C 200 mg/30 mg	2 weekly, 5 fortnightly, 21 monthly
Wu [7]	2010	Taiwan	Retrospective	Single	1985–2007	196 (27/31/138)	Epirubicin 20 mg/20 mL, MMC 10 mg/20 mL	6–8 weekly
O’Brien [9]	2011	UK	Prospective	Multicenter	2000–2006	239 (120/119)	MMC 40 mg/40 mL	Single
Ito [11]	2013	Japan	Prospective	Multicenter	2005–2008	72 (36/36)	Pirarubicin 30 mg/30 mL	Single

Control: No instillation, MMC: Mitocycin-C, Ara-C: Cytosine arabinoside.

**Table 2 jcm-08-01059-t002:** Patient characteristics from eligible studies.

	Median Age, Range (Years)	No. of Gender (M/F)	Tumor Stage (≤T1/T2/T3/T4/NA)	Tumor Grade (High/Low/NA)	Tumor Site (Pelvis/Ureter/Both/NA)	Median FU, Range (Months)
Sakamoto [10]	NA, 55–85	16/9	9/16/0/0/0/	4/21/0	NA	45, 6–65
Wu [7]	65, 23–86	92/104	86/63/47/0/0	81/115/0	54/95/0/37	55.6, 12–182
O’Brien [9]	NA, 36–90	NA	139/32/57/4/7	18/214/7	NA	12
Ito [11]	NA	43/29	39/8/25/0/0	39/33/0	40/28/3/0	24.9, 2.6–39.3

NA: Not available, FU: Follow-up.

**Table 3 jcm-08-01059-t003:** Ranking of intravesical recurrence-free survival outcomes among chemotherapeutic agents.

**All Studies**										
*** Rank**	**A**		**B**		**C**	**D**		**E**		**F**
1	46.0		4.9		2.3	1.2		45.6		0.0
2	29.4		19.4		11.3	7.7		32.3		0.0
3	9.7		32.9		25.7	22.0		9.7		0.2
4	6.5		23.8		30.1	32.9		6.0		0.5
5	5.6		16.9		27.3	33.7		4.7		11.7
6	2.8		2.1		3.3	2.5		1.7		87.6
A. MMC 20 mg + Ara-C 200 mg (maintenance). B. MMC 10 mg (induction). C. Epirubicin 20 mg (induction). D. MMC 40 mg (single instillation). E. Pirarubicin 30 mg (single instillation). F. Control.										
**Prospective Controlled Trials**										
*** Rank**		**A**		**B**			**C**		**D**	
1		49.2		2.3			48.5		0.0	
2		37.9		21.2			40.8		0.2	
3		10.0		74.0			9.0		7.0	
4		2.8		2.6			1.8		92.8	
A. MMC 20 mg + Ara-C 200mg (maintenance). B. MMC 40 mg (single instillation). C. Pirarubicin 30 mg (single instillation). D. Control.										

The values correspond with the probabilities that the drug holds the indicated rank (% of 2000 iterations). * Lower rank indicated a more efficacious for the noted outcome. The overall ranking is determined by the smallest sum of the values multiplied by each rank and its probability.

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
