# Peer review of "Intravesical Chemotherapy after Radical Nephroureterectomy for Primary Upper Tract Urothelial Carcinoma: A Systematic Review and Network Meta-Analysis"

_jcm, 2019, doi:10.3390/jcm8071059_

Round 1
Reviewer 1 Report
JCM – 517216
The authors present a Systematic Review on Intravesical chemotherapy after radical nephroureterectomy for primary upper tract urothelial carcinoma: a systematic review and network meta-analysis.
There are two main characteristic that distinguish a medical systematic review from a comprehensive review. The first one is the identification of a relevant clinical question and second one its transparency.
Regarding the first characteristic, there is a reasonable doubt that their clinical question is relevant since there is only a reasonably good RCT (O’Brian published in 2011) and their conclusions are mainly applicable to non-invasive /Low grade tumours. References 10 and 11 are clearly unpowered to show any difference in terms of the outcome evaluate in this systematic review and reference 7 is a retrospective study. In fact, any level of evidence reached by the conclusions of the present SR is superseded by and does not reach the level of evidence of the RCT of O’Brian.
- The abstract starts by “the aim of this study was to …. by systematic review and network meta-analysis of RCT”. Five lines below in the same abstract it can be read: Three RCTs on multicentre setting and one large retrospective study….” Please clarify.
- Abstract line 25 “On with Ara -C…” do they mean both Ara-C and induction therapy of MMC?
- Point 2.4 describes in detail the multiple statics test used in their meta-analysis. Most of them are only understandable to a statistician and being out of comprehension of the clinicians.
- Without questioning the methodology of the authors, it seems weird that there is no significant heterogeneity among studies specially regarding tumour Stage and Grade distribution but also follow-up.
- It is interesting their Figure 3. This Bayesian framework network metanalysis. Can the authors explain these figures? I’m afraid it is unclear unless you are verse in statistics. Furthermore, the dispersion in terms of treatment drug and duration is large so difficult to reach any conclusion in term of effectiveness of one or another regime.
- Also, table 3, the ranking of intravesical recurrence-free survival outcomes is difficult to follow
- Figure 1 (PRISMA Flow chart) is difficult to follow. Usually the figures or number of records included from bottom to down decrease as a result of a through selection process. In this PRISMA Flow chart that is not the case. The bottom case contains 350 records, the following 144 and the third one 206. Something is missed in this flow diagram.
- Table 1 is incomplete regarding the distribution of groups chemotherapy and control. What control means and how the different groups were distributed in the possible treatment arms should be specified in the table and clarify in the text.
- Risk of bias assessment of the included studies is missed. Unfortunately, risk of bias publication is a totally different matter.
- In the discussion first paragraph, the authors stated that network met-analysis revealed that Pirarubicin was the most effective regimen among intravesical chemotherapy strategies. Though it can be statistically sound to base their results on such a sophisticated Bayesian analysis, the real question is it is clinically sound to base their statement in a group of exclusively 36 patients treated with Pirarubicin (see table I). Do they suggest we should base our clinical practice on the results of a phase II RCT clearly unpowered to show any superiority and not even design as inferiority? While this Bayesian analysis may be hypothesis generating, a SR rarely generates hypothesis.
Author Response
The authors present a Systematic Review on Intravesical chemotherapy after radical nephroureterectomy for primary upper tract urothelial carcinoma: a systematic review and network meta-analysis.
There are two main characteristic that distinguish a medical systematic review from a comprehensive review. The first one is the identification of a relevant clinical question and second one its transparency.
Regarding the first characteristic, there is a reasonable doubt that their clinical question is relevant since there is only a reasonably good RCT (O’Brian published in 2011) and their conclusions are mainly applicable to non-invasive /Low grade tumours. References 10 and 11 are clearly unpowered to show any difference in terms of the outcome evaluate in this systematic review and reference 7 is a retrospective study. In fact, any level of evidence reached by the conclusions of the present SR is superseded by and does not reach the level of evidence of the RCT of O’Brian.
- The abstract starts by “the aim of this study was to …. by systematic review and network meta-analysis of RCT”. Five lines below in the same abstract it can be read: Three RCTs on multicentre setting and one large retrospective study….” Please clarify.
Reply>
Thank you for your sharp comment. We totally agree with you.
This study was initially conducted for RCT only. However, there were too few RCT studies on this subject, and one large-scale retrospective cohort study with a sufficient level of evidence was found. Thus, one retrospective study included in this study with the consensus of the researchers.
We revised the manuscript to make it, as your comment. We apologize for using the indefinite expression.
- Abstract line 25 “On with Ara -C…” do they mean both Ara-C and induction therapy of MMC?
Reply>
It seems that there was a big mistake.
Originally it was the sentence: “On network meta-analysis, pirarubicin was ranked the most effective regimen while maintenance therapy of MMC with Ara-C and induction therapy of MMC was ranked as the second and third most effective regimens, respectively.”
I do not know how this happened. It is thought to be an error caused by the transmission process.
A good point is really grateful. We’ll read this manuscript carefully to see if there are any other errors and fix them all.
- Point 2.4 describes in detail the multiple statics test used in their meta-analysis. Most of them are only understandable to a statistician and being out of the comprehension of the clinicians.
Reply>
Network meta-analysis is a very recent statistical method, and I heard that only some experts among statisticians can implement it. Thus, all clinicians cannot understand these latest analytical techniques, and not all clinician needs to understand this latest statistical method.
Although it would be better to describe for all readers to be able to understand it easily, but it is too difficult statistical technique, so it is not easy to deal with detailed and clarity in this study. It takes too long to deal with this statistical technique in detail. Since our research is not a study of statistical techniques, it does not seem to fit the purpose of the study. Instead, we cited eight references so that interested readers could easily find them.
I am also a urological clinician and I am not familiar with this statistical technique. I do not think I will be able to understand all the statistical contents at such a highly specialized era as these days. This is why collaboration between different fields is important, and we think that this phenomenon will become more intense in the future.
- Without questioning the methodology of the authors, it seems weird that there is no significant heterogeneity among studies specially regarding tumour Stage and Grade distribution but also follow-up.
Reply>
The efficacy of intravesical chemotherapy for urothelial cell carcinoma is already well known and is highly recommended in many guidelines with a high level of evidence.
The purpose of our study is not to confirm the efficacy of intravesical chemotherapy. The aim of this study is to find out which drugs are most effective among effective drugs. Therefore, we chose studies that reported that intravesical chemotherapies were effective, and therefore may have reported similar results.
- It is interesting their Figure 3. This Bayesian framework network metanalysis. Can the authors explain these figures? I’m afraid it is unclear unless you are verse in statistics. Furthermore, the dispersion in terms of treatment drug and duration is large so difficult to reach any conclusion in term of effectiveness of one or another regime.
Reply>
A network meta-analysis is a statistical method that indirectly compares common comparative subjects when there is no direct comparative study of two drugs or direct comparison is difficult. Figure A shows that the A, B, C, D, and E regimens were compared based on the effect of the control (no instillation), which means F node. Figure B shows a comparison of the effects of A, B, and C regimens on control (D node) by collecting only RCT studies with a higher level of evidence. It seems that the explanation was not enough in the manuscript. We have added a more detailed description of this part.
We are very sorry. It also looks like the table did not have enough explanation. The values correspond with the probabilities that the drug holds the indicated rank. However, the overall ranking is determined by the smallest sum of the values multiplied by each rank and its probability. A more detailed description has been added to the caption of the table.
- Figure 1 (PRISMA Flow chart) is difficult to follow. Usually the figures or number of records included from bottom to down decrease as a result of a through a selection process. In this PRISMA Flow chart that is not the case. The bottom case contains 350 records, the following 144 and the third one 206. Something is missed in this flow diagram.
Reply>
It is our mistake. We seem to have misunderstood the PRISMA flow chart. N = 144 in the second column means the number of studies removed. However, it was actually a space to write the number of research that remained after the removal.
Based on your advice, we revised the PRISMA flow chart.
- Table 1 is incomplete regarding the distribution of groups chemotherapy and control. What control means and how the different groups were distributed in the possible treatment arms should be specified in the table and clarify in the text.
Reply>
Thank you for your sharp remark. All studies compared no instillation as control. As you recommend, the control is marked "no instillation" in the text and in Table 1. Three prospective studies were distributed equally by randomization, so there was no significant difference in baseline characteristics between the two groups and it was not specifically mentioned. For one retrospective cohort study, the distribution between groups is clarified in the text.
I appreciate it has become a more readable manuscript through your recommendation.
- Risk of bias assessment of the included studies is missed. Unfortunately, the risk of bias publication is a totally different matter.
Reply>
Thank you for your valuable comment, and we totally agree with your comment.
We have added the risk of bias assessment as figure 2.
- In the discussion first paragraph, the authors stated that network meta-analysis revealed that Pirarubicin was the most effective regimen among intravesical chemotherapy strategies. Though it can be statistically sound to base their results on such a sophisticated Bayesian analysis, the real question is it is clinically sound to base their statement in a group of exclusively 36 patients treated with Pirarubicin (see table I). Do they suggest we should base our clinical practice on the results of a phase II RCT clearly unpowered to show any superiority and not even design as inferiority? While this Bayesian analysis may be hypothesis generating, an SR rarely generates hypothesis.
Reply>
We totally agree with your opinion. Our study does not offer conclusions that change the practical pattern. We hope our study to have meaning just to summarize the evidence to date.
The theme of this study is a very difficult subject to implement RCT. In addition, because new drugs are not available in this field, it seems that the clinical trial is not actively progressing. This is why we conducted this network meta-analysis study.
We hope that more good research will be carried out in the future, and a clearer conclusion can be reached. It is our hope that our research would make a small contribution to such efforts.
Reviewer 2 Report
This is a nice review work. However, this review is performed mostly on papers published 5-years ago. Could the authors perform review on more recent works?
Author Response
Thank you very much for your comment. I totally agree with your opinion.
In this study, the date of literature search is December 22, 2016. We found 350 articles in three of the most popular databases, and the most recent study was found in 2013.
The theme of this study is a very difficult subject to implement RCT. In addition, because new drugs are not available in this field, it seems that the clinical trial is not actively progressing. This is why we conducted this study.
At this point, we did a search again, but the well-conducted RCT studies were not retrieved newly. Our study is now considered to be the most up-to-date evidence of medical consequences of this field of study.
Reviewer 3 Report
Comments:
The authors indicate that “Intravesical chemotherapy after radical nephroureterectomy for primary upper tract urothelial carcinoma: a systematic review and network meta-analysis”. I have few minor comments.
1. Overall, the review article submitted by the authors are interesting and done with good effort.
2. Line 83 Intravesical –recurrence fee
3. At the end only n = 4 studies are included in their meta-analysis. They should have included more than n = 350 studies in their database searching.
4. Line 230, In Intravesical chemotherapy has been proven to be effective (need reference)
Author Response
1. Overall, the review article submitted by the authors are interesting and done with good effort.
Reply>
Thank you very much for your warm comment.
2. Line 83 Intravesical –recurrence fee
Reply>
Thank you for your sharp comment. It is our big mistake. We apologize for our carelessness.
We revised the manuscript to “recurrence free” as your comment.
3. At the end only n = 4 studies are included in their meta-analysis. They should have included more than n = 350 studies in their database searching.
Reply>
Thank you for your comment. We also were a little surprised at the very small number of studies.
In this study, the date of the literature search is December 22, 2016. We found 350 articles in three of the most popular databases, and the most recent study was found in 2013.
The theme of this study is a very difficult subject to implement RCT. In addition, because new drugs are not available in this field, it seems that the clinical trial is not actively progressing. This is why we conducted this meta-analysis study.
At this point, we did a search again, but the well-conducted RCT studies were not retrieved newly. Our study is now considered to be the most up-to-date evidence of the medical consequences of this field of study.
4. Line 230, In Intravesical chemotherapy has been proven to be effective (need reference).
Reply>
Thank you for your valuable comment. We totally agree with your comment.
The effectiveness of intravesical chemotherapy is well known, and it is recommended as a high-level recommendation as a standard treatment in most urologic guidelines. We cited some of the most representative guidelines of urology as you recommend.
1. Rouprêt, M.; Babjuk, M.; Burger, M.; Compérat, E.; Cowan, N.C.; Gontero, P.; Mostafid, A.H.; Palou, J.; van Rhijn, B.W.G.; Shariat, S.F., et al. EAU Guidelines on Upper Urinary Tract Urothelial Carcinoma 2018. In European Association of Urology Guidleines. 2018 Edition., European Association of Urology Guidelines Office: Arnhem, The Netherlands, 2018; Vol. presented at the EAU Annual Congress Copenhagen 2018.
2. Chang, S.S.; Boorjian, S.A.; Chou, R.; Clark, P.E.; Daneshmand, S.; Konety, B.R.; Pruthi, R.; Quale, D.Z.; Ritch, C.R.; Seigne, J.D., et al. Diagnosis and Treatment of Non-Muscle Invasive Bladder Cancer: AUA/SUO Guideline. The Journal of urology 2016, 196, 1021-1029, doi:10.1016/j.juro.2016.06.049.